# Clinical Profiles and Factors Associated with a Low Sodium Intake in the Population: An Analysis of the Swiss Survey on Salt

**DOI:** 10.3390/nu12113591

**Published:** 2020-11-23

**Authors:** Michel Burnier, Fred M. Paccaud, Murielle Bochud

**Affiliations:** 1Service of Nephrology and Hypertension, University Hospital, 1011 Lausanne, Switzerland; 2Hypertension Research Foundation, St. Légier, 1011 Lausanne, Switzerland; 3Center for Primary Care and Public Health-Unisanté, Department of Epidemiology and Health Systems, University of Lausanne, 1015 Lausanne, Switzerland; fred.paccaud@gmail.com (F.M.P.); murielle.bochud@unisante.ch (M.B.)

**Keywords:** sodium intake, cardiovascular risk, blood pressure, gender, protein intake, meat consumption

## Abstract

As a part of the salt controversy, it has been suggested that people with a low sodium intake have an increased risk of cardiovascular events. However, there is no clear explanation for this increased risk. We examined the socio-demographic, clinical profile, and behavioral factors associated with a low sodium intake in the Swiss subjects who participated in the Swiss Survey on Salt. Only 13.3% of the Swiss population eat less than 5 g of salt daily and among them 78.2% are women. Subjects with a low sodium intake eat and drink less as reflected by lower intakes of proteins, potassium, and calcium and a smaller urine volume. In addition, a low blood pressure, a normal body mass index, a low prevalence of obesity, a low serum uric acid, and less alcohol and cigarette consumption characterized this group, suggesting a rather low cardiovascular risk profile. Being single and doing most of the cooking at home are associated with a low intake of sodium, as well as a less frequent consumption of meat and fish when eating less than 5 g salt per day. However, the awareness of the effects of salt on health and cardiovascular risk, health concerns, and physical activity are similar in subjects eating more or less salt. In conclusion, we could not evidence clinical or behavioral factors that could significantly increase the risk of developing cardiovascular events in low salt eaters.

## 1. Introduction

Organizations such as the World Health Organization (WHO) [1], the Institute of Medicine in the United States [2], and several scientific societies [3,4] recommend that adults eat no more than 5-6 g of salt (NaCl) per day (or <2000 to 2400 mg Na/day) to prevent cardiovascular diseases. Yet, salt intake remains much higher than recommended in most countries around the world, and this might be the cause of millions of deaths from cardiovascular diseases [5]. Whereas the cardiovascular burden of a very high sodium diet is well documented and recognized by the scientific community, the potential benefits of lowering daily salt intake to ≤5 g/day remain highly controversial. Indeed, several prospective observational studies have reported that subjects with a low baseline dietary sodium intake have a higher all-cause and cardiovascular mortality during follow-up [6,7,8]. These observations, confirmed by some meta-analyses, suggest that people with a low sodium intake have an increased risk of cardiovascular events almost comparable to those on a high salt diet [9]. Yet, other observational and interventional studies do not support this association and rather report a linear relation between sodium intake and the risk of cardiovascular morbidity and mortality [10,11,12,13,14]. For the authors of these findings, the increased risk associated with a low sodium intake could be due to methodological issues, linked to the assessment of sodium intake, and/or to reverse causality [15,16,17]. Another potential explanation for the J-shape association between salt consumption and cardiovascular events could be that a percentage of subjects on a low salt intake were characterized by an inverse salt-sensitivity leading to an increase in blood pressure on a low salt diet [18].

In observational studies reporting a higher cardiovascular risk and mortality on a salt intake below 7 g NaCl per day (or a Na intake below 3000 mg/day), the authors provide few details on the clinical and behavioral characteristics of participants composing this subgroup [7,8,19]. Thus, before extrapolating their conclusions to the general population and warning against the detrimental effects of a low salt diet, it is important to better characterize the clinical and behavioral features of people on a low salt intake. This may enable a better understanding of why these people might be (or might not be) at high risk of developing cardiovascular events and dying from a cardiovascular cause within a relatively short period of observation. In the present paper, we address this question by exploring data from the Swiss Survey on Salt, a population-based survey conducted between 2010 and 2012. We examined the socio-demographic and clinical profile and the behavioral factors associated with a low sodium intake in the Swiss population aged 15-year and over, comparing participants with a sodium intake below 5 g NaCl per day with those consuming 5 g NaCl per day or more based on the 24-h urinary excretion of sodium.

## 2. Materials and Methods

The Swiss Survey on Salt was a cross-sectional, population-based survey conducted in 9 out of 26 cantons (Vaud, Geneva, Valais, Fribourg, Luzern, Basel, Zürich, St. Gallen, and Ticino), which covered the three main linguistic regions of Switzerland. This survey was part of a nationwide program of the Swiss Federal Office of Public Health that aimed at reducing salt intake in the population. To be included, participants had to be permanent residents of Switzerland and aged 15 years or older. Recruitment took place from January 2010 to March 2012 and was based on a two-level sampling strategy similar to the one used for the Swiss Health National Surveys conducted every five years [20]. The Swiss Statistical Office provided a list of randomly selected households from the Swisscom fixed-line phone directory, updated every 3 months, separately for each canton (Swisscom is a major telecommunication provider in Switzerland, partially dependent from the federal state). This directory is the largest and most complete directory of the country and covers most Swiss households.

In a first step, we contacted households by sending an information letter followed by phone calls with a maximum of three attempts on different days. In a second step, we randomly selected, during the phone call, one person from the household to take part in the study, using a computer-generated random number. We recruited participants in eight predefined sex- and age-strata (men and women aged 15–29, 30–44, 45–59, and ≥60 years). Because of difficulties recruiting young and middle-aged people, a limited convenience sampling (defined as a non-probability sampling involving subjects being drawn from the population that is close to hand) was used in selected centers. Thus, out of 1515 who completed the survey, we recruited 277 participants (18%) by way of convenience sampling. The overall participation rate was 10%. Participants from the Italian region were slightly overrepresented.

### 2.1. Data Collection

The main goal of the study was to estimate dietary salt intake in the Swiss adult population based on a single 24-h urine collection. In order to avoid a change in usual salt intake before the urine collection, we told participants that the study was about “lifestyle and blood pressure” and specific care was taken not to mention salt intake at the time of enrolment. The study participants attended the study center on two occasions: one baseline visit before and one follow-up visit after completion of the urine collection (within 24 h).

At the baseline visit, participants filled in a questionnaire about socio-demographic factors, medical history, lifestyle, and medication. Questions about alcohol intake were asked, including units of wine, beer, and spirits consumed in the past 7 days. A standard unit was defined as a glass of wine, a bottle of beer or a shot of spirits, approximating 10–12 g ethanol [18]. We classified participants into three mutually exclusive categories according to the total number of alcohol units reported in the past 7 days (0, 1–6, ≥7 units per week). Smoking status was classified as current smoker versus non-smoker.

We measured weight in light indoor clothing to the nearest 100 g with a medical scale and height was measured to the nearest centimeter with a wall-mounted stadiometer. Body mass index (BMI) was calculated as weight divided by height in meters squared. Overweight was defined as BMI ≤25 kg/m^2^ and <30 kg/m^2^, and obesity as BMI ≥30 kg/m^2^. Blood pressure was measured at least five times at each visit in the left arm after at least 5 min rest in the seated position, with a clinically validated automated oscillometric device (Omron^®^ HEM-907, Matsusaka, Japan) with an adapted standard cuff [21]. Hypertension was defined as mean systolic blood pressure ≥140 mm Hg or mean diastolic blood pressure ≥90 mm Hg or use of antihypertensive medication.

We gave participants two 3-litre plastic bottles and standard instructions on how to collect a 24-h urine specimen. When brought back at the second visit, urine was weighed, mixed, sampled in small aliquots, and immediately frozen (−20 °C) at each study center. During the second visit, an optional non-fasting blood sample was collected. Urine and blood samples were subsequently sent to the Central Chemical Laboratory of Lausanne University Hospital (CHUV, Lausanne, Vaud, Switzerland) for centralized analyses with use of standard methods and stringent internal quality controls. This Central Chemical Laboratory is ISO/CEI 17025 and ISO 15189 accredited and is regularly checked by the “Swiss Centre for Quality Control”.

We measured plasma and urinary electrolytes by means of indirect potentiometry; urine urea was measured with the urease-GLDH method, and serum and urinary creatinine with the Jaffé kinetic compensated method. We measured Na^+^ excretion in 24-h urine and used the following conversion factor to estimate 24-h urinary salt excretion: 1 mmol of Na^+^ corresponds to 0.0584 g of salt (NaCl). Protein intake was estimated using the Maroni formula, based on measured 24-h urinary urea excretion [22,23]. We calculated the estimated glomerular filtration rate (eGFR) using the Chronic Kidney Disease-Epidemiology Collaboration (CKD-EPI) formula.

We verified the quality of 24-h urine collections from the urinary creatinine excretion adapted for sex and weight. The reproducibility of urinary Na^+^ excretion was verified in a small subset of participants (*n* = 57) who made two collections. The coefficient of correlation between the two collections was 0.69 for Na^+^ excretion and 0.71 for urinary volume. We excluded 97 subjects from the analysis because of incomplete urine collection. Criteria for exclusion were a urinary volume less than 300 mL/24 h (*n* = 3), participants reporting not having collected all 24-h urine (*n* = 29), and a urinary creatinine excretion equal to or below the sex-specific fifth percentile: 0.121 mmol/kg/24 h in men and 0.082 mmol/kg/24 h in women (*n* = 77).

At the end of the second visit, participants answered the following questions:-“Do you consider that a diet rich in salt has an impact on your health” (possible answers: yes, no, do not know)-“According to you, for which diseases is there a direct link with salt intake?” (arterial hypertension, cardiac diseases, diabetes, irritable bowel syndrome, no disease, over-weight/obesity, myocardial infarction, stroke, tuberculosis, other)-“Which type of salt do you use at home?” (salt without supplements, iodized salt, salt enriched with iodine and fluoride, spicy salt, do not know)-“Do you usually try to limit your dietary salt intake?” (yes/no)-“Are you usually interested in your daily dietary salt intake?” (yes/no)-“What is the maximum amount of salt a Swiss adult should eat on a daily basis?” (5 g, 7.5 g, 10 g, 15 g, do not know)-“Do you usually add salt to your meals at home / outside of home?” (always, usually, occasionally, rarely)-“Rank the following items with respect to their salt content, taking portion size into account” (one teaspoon of dressing sauce (15 g), one medium-sized bowl of soup (300 mL), one medium-sized slice of bread (30 g), one teaspoon of Knorr Aromat (19 g));-“What is the main source of salt in the daily diet of a standard person living in Switzerland?” (salt added to the meal at the table, fast foods or processed meals, mineral waters, breads, meat and cold cuts, cheeses, soups, do not know).

### 2.2. Statistical Analyses

Sample size calculation was based on the primary outcome of interest, namely dietary salt intake, with the aim of being able to detect (with 80% power at a 5% two-sided type 1 error rate) a 10% difference in salt intake in each sex strata in the four pre-specified age groups and in the two larger linguistic regions (German vs. French). In the present analysis, urinary sodium excretion was separated in two groups, i.e., ≥5 g NaCl/day or <5 g NaCl, corresponding to a 24-h urinary sodium excretion ≥85 mmol Na/24 h or <85 mmol Na/24 h. This cutoff was based on the WHO recommendations for salt intake and corresponds to a sodium intake above or below 2000 mg/day according to American recommendations.

We used means and standard deviation to describe unadjusted continuous variables, and percentages to describe dichotomous or categorical variables. Prevalence was described across sex and age groups. We used analysis of variance (ANOVA) and the chi-squared test to compare the characteristics of groups where appropriate. We conducted multiple logistic regression with the level of sodium intake as the dependent variable, including linguistic region, age, sex, smoking status, alcohol consumption, BMI, eGFR, serum urea, serum uric acid, serum Na^+^, serum K^+^, corrected serum Ca^++^, urinary excretion (Na^+^, K^+^, urea, creatinine), and urine volume as covariates. We conducted multiple linear regression with the level of sodium intake as the dependent variable, including linguistic region, age, sex, smoking status, alcohol consumption, BMI, eGFR, serum urea, serum uric acid, serum Na^+^, serum K^+^, urinary excretion (Na^+^, K^+^, urea, creatinine), and urinary volume. To ease interpretation of the findings, we used sex-specific tertiles of serum and urine Na^+^, K^+^, and urea instead of continuous covariates. For both models, we used a backward procedure, starting with a full model, and then removing, one at a time, non-significant covariates. We forced age, sex, BMI, urine volume, duration of urine collection, and urinary creatinine excretion into the models. The choice of covariates was based on clinical knowledge and previous research, as well as an interest in exploring the available nutrition biomarkers.

## 3. Results

As presented in Figure 1 showing the distribution of sodium intake in the studied population, only 193 out of the 1448 participants, i.e., 13.3%, had a daily salt intake <5 g NaCl. The low sodium pattern was more frequent among women than men. Table 1 and Table 2 present the socio-demographic and lifestyle variables (Table 1) and the anthropometric and biological characteristics (Table 2) of the participants in the two sodium intake groups. In the bivariable analyses, marital status and living alone were associated with a lower sodium intake and a more frequent alcohol consumption was associated with a higher likelihood of eating more than 5 g of salt per day.

Table 2 shows that 78.2% of people eating less than 5 g of salt per day were women. Accordingly, they were leaner, had a lower blood pressure, excreted less creatinine in the urine, but had a similar glomerular filtration rate than those eating more than 5 g salt per day. The prevalence of obesity was also significantly lower in this subgroup (3.6% versus 15.4%, *p <* 0.001). In addition, they had lower serum urea and uric acid levels and lower 24-h urinary urea and potassium excretion suggesting a lower protein and potassium intake. Urinary volume was significantly lower in the <5 g NaCl group by almost 500 mL/day (*p <* 0.001). Regarding hypertension, the prevalence (based on a mean of 8 measurements ≥140/90 mmHg) was 23.0% in the less than 5 g of salt per day group and 26.5% in the group eating more than 5 g of salt daily (*p* = 0.77, ns). The prevalence of antihypertensive therapy was 15% and 15.8%, respectively in the low salt and high salt eaters (*p* = 0.37, ns).

Table 3 summarizes the significant determinants of eating less than 5 g of salt per day in our population in the multivariable logistic regression. Being a woman was a strong determinant associated with a high likelihood of eating less than 5 g of salt per day. Among other determinants of eating less than 5 g of salt per day were a lower estimated protein intake and a lower urinary volume. Being obese was a significant determinant of a high sodium intake only when data were not corrected for protein intake. Linguistic region, eGFR (CKD-EPI), serum uric acid level, nationality, self-reported physical activity, marital status, education level, alcohol consumption, and urinary K excretion were not significantly associated with eating less than 5 g of salt par day in the multivariate analysis.

Table 4 shows the answers to the questionnaire on food intake and preferences and physical activity. In the bivariable analysis, the group of subjects eating less than 5 g of salt per day had a more regular consumption of fruits and a lower consumption of caffeine, fluid, meat, and fish. The amount of liquid reported per day was significantly positively associated with eating less than 5 g of salt per day (OR = 1.60 (1.18-2.27), *p* = 0.01). Yet, neither fruit nor vegetable consumption was a significant determinant of eating less than 5 g of salt per day in a multivariable model. There was also no association of caffeine consumption, physical activity, paying attention to diet, or following a diet within the past 12 months with low salt intake.

People who ate less than 5 g of salt per day ate little protein from meat and from fish. Indeed, a frequent consumption of meat was associated with lower odds of eating less than 5 g of salt per day, but only when protein intake was not included in the model. Results were similar for the number of days eating fish. Both were significant in the multivariable model without estimated protein intake. Figure 2 shows the probability of eating less than 5 g of salt per day according to the reported frequency of eating meat.

Being the person doing most of the cooking at home tended to be associated with a higher probability of eating less than 5 g of salt per day. Indeed, doing most of cooking at home was significantly more frequent in subjects eating a low sodium diet (73.39%) than in those on a higher sodium intake (58.8%, *p <* 0.001). However, the difference was not significant in the multivariate analysis.

Appendix A shows the proportion of participants who correctly classified their dietary salt intake in the corresponding sex-specific quintiles of 24-h urinary salt excretion. Proportions of respondents who estimated their own salt consumption within the corresponding quintile were 33.3% (95% CI: 24.0–43.7), 20.9% (16.9–25.5), 20.3% (17.6–23.3), 21.4% (14.6–29.7), and 46.2% (19.2–74.9), respectively, in each self-reported salt consumption category from very low to very high. In our analysis, reporting to eat high amounts of salt was strongly negatively associated with eating less than 5 g of salt per day (OR = 0.20 (0.07; 0.61); *p* = 0.005). This suggests that when people eat small amounts of salt, they do not always realize that they do so, whereas when people think that they eat a lot of salt, they are more likely to do so. As shown in Appendix A, people who eat less than 5 g of salt per day did not have a better knowledge of health issues related to high salt intake than those eating more salt.

As most participants eating less than 5 g of salt per day were women, we performed additional analyses comparing only women eating more or less than 5 of salt per day. Results are presented in Appendix A. Women from the low salt intake group differed significantly from those eating a higher salt intake on a few parameters. The low intake group had a lower BMI and a lower consumption of alcohol, caffeine, proteins, and potassium and had a small 24-h urine volume. There were no differences between the groups in terms of blood pressure and physical activity.

## 4. Discussion

The objective of our analyses was to examine the clinical profile and the factors associated with a low sodium intake in the Swiss population aged 15 years and over. We showed that only 13.3% of the Swiss population eats less than 5 g of salt per day (<2000 mg Na/day) and among them 78.2% are women. Subjects with a low sodium intake eat and drink less in general as reflected by lower intakes of proteins, potassium, and calcium and a smaller urine volume. Subjects of the low sodium group have a low blood pressure, a normal body mass index, a low prevalence of obesity, a low serum uric acid, and less alcohol and cigarette consumption, suggesting a rather low cardiovascular risk profile. Answers to the questionnaire indicate that the marital status (mainly being a single) and doing most of the cooking at home are associated with a low sodium intake and that meat and fish consumption is significantly less frequent in subjects eating less than 5 g of salt per day. However, the awareness of the effects of salt on health and cardiovascular risk, health concerns, and physical activity are similar in subjects eating more or less salt.

In this Swiss survey, the percentage of participants eating less than 5 g of salt per day is low, with a marked difference between women and men as illustrated in Figure 1. Although age was not a major determinant of salt consumption in our analyses, the percentage of subjects on low sodium is slightly higher in younger women (20.9%) and women older than 60 years (29%). These figures are in accordance with those of similar surveys conducted in the United States [24,25]. Thus, in the 2011–2016 National Health and Nutrition Examination Survey (NHANES), the adherence to US Department of Agriculture Dietary sodium recommendations (<2300 mg Na/day or <6 g NaCl/day) was slightly above 20% in the overall population. In all subjects with a high cardiovascular risk such as hypertensive patients and patients with diabetes or chronic kidney diseases, it was below 10% [24]. The significantly higher representation of women among subjects eating less than 5 g of salt is not surprising as women were found in all surveys including ours to consume less salt than men [26]. In the two large surveys published by O’Donnell et al. [7] and by Stolarz-Skrzypek et al. [8] demonstrating an increased cardiovascular risk associated with a low sodium intake, the percentage of women in the low salt groups was, respectively, 52.8% and 70.4%. However, their cut-offs of salt excretion were higher. Indeed, the upper cut-off of the low salt group was <7.6 g NaCl/day in O’Donnell’s survey and <7.4 g NaCl/day in women and <9.3 g NaCl/day in men in Stolarz-Skrzypek’s study. These differences may have a significant influence on the estimation of the cardiovascular risk and the occurrence of cardiovascular events.

As mentioned earlier, one reason to perform these analyses was to assess in more detail the demographic, clinical, and behavioral characteristics of subjects eating salt according to WHO recommendations [1] in a population that might be in part representative of Western Europe due to its diversity, as it represents subjects of French-, German- and Italian-speaking regions. When compared to subjects eating 5 g of salt per day or more, several differences were observed. Firstly, our data indicate that subjects with a low urinary sodium excretion eat less food as reflected by a significantly lower intake of proteins, potassium, and calcium. This may be the reason why these subjects have a lower BMI and a low prevalence of obesity. In terms of food choice, these subjects eat meat, fish, and more fruits less frequently and not significantly more vegetables based on frequency. This observation was still present when comparing only women on a low or high sodium intake. Today, processed meat products contribute to about 20% of the total sodium dietary intake in Western countries. In our survey, a frequent consumption of meat is associated with an increased likelihood of eating more than 5 g salt per day. Secondly, subjects eating less than 5 g salt per day have a smaller urine volume, suggesting that their fluid intake is also lower. This finding is also consistent across analyses. Salt consumption is known to trigger thirst and hence to increase urinary volume. Thus, 24-h urinary volume closely relates to 24-h urinary sodium excretion [26]. In this respect, He et al. have calculated that a 100-mmol/day reduction in sodium intake would decrease 24-h urine volume by 379 and 399 mL in hypertensives and normotensives, respectively [27]. Our data are in accordance with this estimation, as the difference in sodium between the two groups was 6.2 g NaCl per day (106 mmol/day) and the difference in urinary volume was 494 mL per day. The smaller urinary volume could have been due to the higher proportion of women in the low sodium group as fluid intake is generally lower in women than in men [28], although this was not the case in the overall analysis of our Swiss population [29]. Thirdly, two observations were made on the entire population and not when comparing women of the two groups: a significantly lower blood pressure in subjects eating less than 5 g of salt and an increased prevalence of subjects living alone in the household and doing most of the cooking at home. These findings confirm the association between blood pressure and salt intake [19] and demonstrate the impact of social factors on salt intake. Of note, the prevalence of hypertension and treated hypertension was not different in the low and high salt groups. Finally, the questionnaire on behavioral aspects shows that subjects eating less than 5 g of salt per day are not more health conscious than their counterparts are. Indeed, they do not pay more attention to their diet, they have the same level of knowledge about the risks associated with a high sodium intake, and they do not have more physical activity than those eating more salt. These observations would suggest that health concerns and prevention of cardiovascular diseases are not necessarily the driving force for eating less salt. Other factors such as weight control or socio-economic issues, such as poverty or social isolation, might also play an important role leading to an intentional restriction of sodium intake.

One important question is whether these differences could affect negatively the cardiovascular risk of these subjects and, hence, explain the increased cardiovascular morbidity and mortality associated with a low sodium intake in some surveys. At first glance, the association of a high percentage of women with a low blood pressure, a normal body weight, a low prevalence of obesity, a low uric acid content, and a low prevalence of smoking and alcohol consumption would suggest that this group of subjects has a rather low cardiovascular risk profile. Thus, according to the European High Risk Chart calculator of the European Society of Cardiology and considering that Switzerland belongs to the low cardiovascular risk countries, most of these subjects would have a 10-year risk of fatal cardiovascular disease below 1%, whatever their cholesterol levels [30]. In our survey, cholesterol levels were unfortunately not measured. However, a similar Swiss population-based survey including more than 6000 random subjects aged 35–75 years from the Lausanne area in 2012, showed mean cholesterol levels at 5.69 mmol/L, low-density lipoprotein (LDL) levels at 3.45 mmol/L, and high density lipoprotein (HDL) levels at 1.65 mmol/L [31].

One characteristic of our low salt subjects that could affect their risk of developing cardiovascular events is the low protein intake. Protein intake, especially from meat, has been associated with a higher risk of developing cardiovascular events, though results of interventional studies remain controversial [32,33]. In the Prevention of Renal and Vascular ENd-stage Disease (PREVEND) study performed in the Netherlands, a U-shaped association between protein intake and cardiovascular events was observed, suggesting that individuals with either a high or a low protein intake have higher cardiovascular event rates [34]. The association of a high incidence of all-cause and of non-cardiovascular mortality with a low protein intake was particularly surprising. As a possible explanation, the authors of PREVEND have hypothesized that the observed effects of low protein intake were likely due to malnutrition induced by poor health status of this group [35]. The same reason may have accounted for the higher morbidity and mortality observed in O’Donnell’s study [7], which included several countries with a high prevalence of malnutrition [7].

At last, our subjects with a low sodium intake were more likely to live alone. Loneliness and social isolation have been reported to be associated with an increased risk of stroke and cardiac events [36]. How much these social factors may counterbalance the benefits of a low salt intake and contribute to increase the cardiovascular risk of our subjects eating less than 5 g of salt per day is unknown.

Our study has several limitations. The first is that the estimation of sodium intake was based on a single 24-h urine collection. Today’s recommendations are to perform several (at least 3) non-consecutive 24-h urine collections to assess sodium intake more accurately [37]. The second is that some risk factors were not measured, such as glucose, serum cholesterol, or inflammatory markers, which could have improved our estimation of the subjects’ cardiovascular risk. Regarding blood pressure, one cannot exclude that some of our participants with a low salt intake had an inverse salt sensitivity, characterized by an increase in blood pressure on a low sodium diet. This pattern has been reported to occur in about 10% of the normal population [18,38]. However, it has been described essentially in short-term low/high sodium protocols, and whether blood pressure remains high on a low sodium intake in the long-term has not been demonstrated. Moreover, the impact of the inverse salt sensitivity pattern on cardiovascular health has been poorly studied so far. Yet, this pattern could contribute to increase the cardiovascular risk of some subjects on a low sodium intake. In addition, the assessment of socio-economic and nutritional aspects was not exhaustive. Finally, neither mortality nor cardiovascular events were available in this population-based study.

## 5. Conclusions

Taken together, our analyses suggest that Swiss subjects eating less than 5 g of salt per day differ from those eating more salt in several clinical, social, and behavioral aspects. In these subjects, we could not evidence factors that would significantly and indirectly increase their risk of developing cardiovascular events. Quite the contrary, they appear to have a low cardiovascular risk profile. Thus, in the population under study, classical risk factors do not play a role as intermediate factors between low salt consumption and cardiovascular events. There are no arguments suggesting that increasing the salt intake among those eating less than 5 g of salt per day will modify their cardiovascular risk. Hence, one should be careful in extrapolating the results of large international surveys like the Prospective Urban Rural Epidemiology (PURE) survey suggesting that a low sodium intake, as recommended by the WHO, increases cardiovascular morbidity and mortality. This might be true in low-income countries where malnutrition is highly prevalent and may induce a reverse causality. This is unlikely in higher-income countries such as Switzerland. More studies are needed to explore possible negative impacts of a low sodium diet in high-income countries. In the meantime, current recommendations should remain unchanged.

## Figures and Tables

**Figure 1 nutrients-12-03591-f001:**
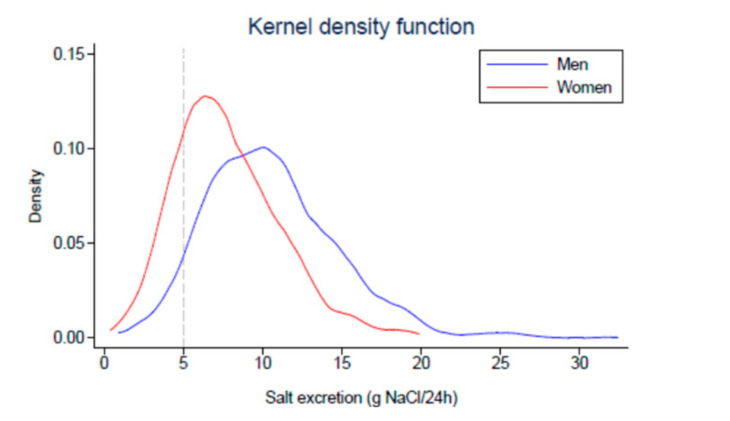
Urinary salt excretion (g NaCl/24 h) distribution according to sex.

**Figure 2 nutrients-12-03591-f002:**
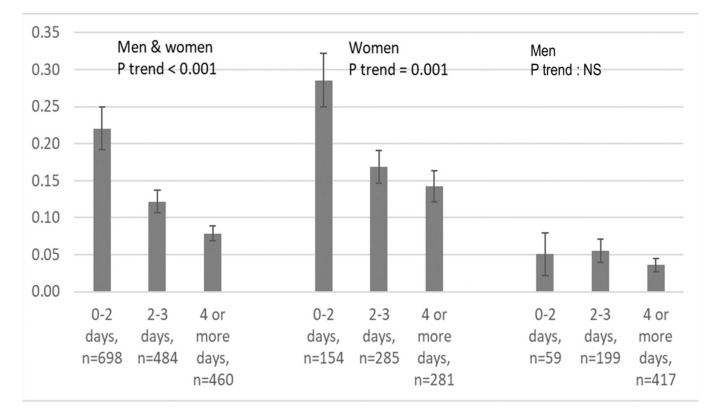
Probability of eating less than 5 g of salt per day according to the reported frequency of eating meat, overall and by sex.

**Table 1 nutrients-12-03591-t001:** Socio-demographic and lifestyle variables according to the level of NaCl intake (more or less than 5 g per day) in the Swiss population.

Variables	≥5 g/24 h (*n* = 1255)	<5 g/24 h (*n* = 193)	*p* (χ^2^)
Civil status (%)			0.002
Single	36.1	38.3	
Married	47.6	36.3	
Other * divorced/separated	16.3	25.4	
Nationality (% Swiss)	86.5	89.1	0.308
Born in Switzerland (% yes)	78.0	82.9	0.122
Linguistic region (%)			0.093
French	30.0	37.8	
German	55.9	49.2	
Italian	14.1	13.0	
Education level (%)			0.478
Low (mandatory or less)	15.5	16.6	
Medium (apprenticeship/high school)	43.7	47.2	
High (university, specialist training, high specialist training)	40.9	36.3	
Living alone in household (% yes)	79.9	69.4	0.001
Current smoker (% yes)	17.6	14.5	0.288
Current alcohol use (% yes)	84.9	72.5	<0.001
Alcohol consumption, frequency (%)			0.003
Never	13.9	24.3	
Less than once per week	26.2	27.0	
1–2 times per week	31.0	28.0	
<1×/day, more than 2×/week	11.8	7.9	
at least once per day	17.1	12.7	
Physical activity level			0.797
Nearly never	16.5	19.2	
Less than once per week	15.0	13.5	
Once per week	19.1	19.2	
More than once per week	49.4	48.2	

* Other means divorced, separated, or widowed.

**Table 2 nutrients-12-03591-t002:** Participants’ anthropometric and biological characteristics by salt intake category (less than 5 g per 24 h vs. ≥5 g/24 h).

Variable	≥5 g/24 h		<5 g/24 h		*p* Value
	*N*	Mean (SD)	*N*	Mean (SD)	
Urine Na excretion (g/24 h)	1255	10.0 (3.6)	193	3.8 (1.0)	<0.001
Age (years)	1255	46.7 (17.9)	193	48.1 (20.9)	0.40
Sex (% women)	1255	47.3	193	78.2	<0.001
BMI (kg/m^2^)	1251	25.5 (4.6)	193	23.0 (3.8)	<0.001
Obesity prevalence (%)	1251	15.4	193	3.6	<0.001
Body weight (Kg)	1251	74.4 (15.7)	193	63.9 (11.7)	<0.001
Body height (cm)	1251	171 (9)	193	167 (8)	<0.001
Menopause (%), (women only)	578	39.3	148	47.3	0.08
Contraceptive pill (%) (premenopausal women only)	342	31.0	76	34.2	0.59
Systolic BP (mm Hg)	1253	124 (15)	192	119 (16)	<0.001
Diastolic BP (mm Hg)	1253	74(10.2)	192	71(8.9)	<0.001
Heart rate (b/min)	1255	71.5 (11.6)	193	74.5 (11.8)	<0.001
Serum creatinine	1147	80.4 (17.8)	169	75.9 (14.8)	0.002
Serum K	1148	4.2 (0.4)	169	4.1 (0.4)	0.22
Serum Na	1148	141.9 (2.0)	169	141.7 (2.4)	0.16
Serum Ca	1148	2.29 (0.10)	169	2.28 (0.10)	0.38
Serum protein	1145	71.3 (4.2)	169	71.2 (5.0)	0.66
Serum urea	1146	5.60 (1.87)	169	5.03 (1.79)	<0.001
eGFR using CKD-EPI	1147	90.4 (18.9)	169	87.7 (22.3)	0.10
Serum uric acid	1148	308 (84)	169	282 (83)	<0.001
Urine urea excretion (mmol/24 h)	1255	388 (135)	193	237 (95)	0.002
Estimated protein intake (g/24 h)	1251	82.3 (25.2)	193	53.8 (17.5)	<0.001
Urine K excretion (mmol/24 h)	1255	69.6 (24.4)	193	46.9 (20.3)	<0.001
Urine creatinine excretion (mmol/kg/24 h)	125	0.18 (0.05)	193	0.14 (0.05)	<0.001
Urine volume (mL/24 h)	125	2016 (910)	193	1522 (809)	<0.001

BMI: body mass index; BP: blood pressure; eGFR: estimated glomerular filtration rate; CKD-EPI: chronic kidney disease Epidemiology formula.

**Table 3 nutrients-12-03591-t003:** Multivariable logistic regression of eating less than 5 g/24 h of salt.

*N* = 1396	OR	95%CI	*p* Value
Age, years	1.005	0.993;1.015	0.418
Sex (being women)	1.75	1.03;2.99	0.039
Birth place (Switzerland)	2.10	1.23;3.61	0.007
BMI < 25	1 (ref)		
Overweight	0.91	0.557;1.49	0.715
Obesity	0.43	0.154;1.23	0.116
Urinary creatinine (mmol/kg/24 h) *	0.700	0.002; 266	0.906
Estimated protein intake (10 g/day)	0.56	0.47;0.66	<0.001
Urinary Ca excretion (mmol/24 h)	0.88	0.78;0.99	0.048
Urine volume (L/24 h)	0.59	0.46;0.75	<0.001

Age, sex, BMI categories, urine volume, and urinary creatinine excretion were forced into the model. The other variables needed to have *p* < 0.10 to stay in the model. Linguistic region, eGFR (CKD-EPI), serum uric acid level, nationality, self-reported physical activity, marital status, education level, alcohol consumption, and urinary K excretion were not significantly associated with eating less than 5 g of salt par day. * square-root transformed.

**Table 4 nutrients-12-03591-t004:** Behavioral characteristics in the overall population (questionnaire data) according to 24-h salt intake (more or less than 5 g NaCl per day).

	*N*	≥5 g/24 h	<5 g/24 h	*p* Value
		%	%	
Consumption of fruits	1520			0.03
Less than once per day		28.60	23.15	
1–2 portions per day		50.15	48.15	
3 or more portions per day		21.2	28.70	
Consumption of vegetables	1527			0.21
Less than once per day		22.35	24.54	
1–2 portions per day		57.13	50.93	
3 or more portions per day		20.52	24.54	
Doing most of cooking at home	1538	58.86	73.39	<0.001
Caffeine, %, 4 or more cups/d	1537	27.10	18.52	0.008
Paying attention to diet	1536	66.46	70.23	0.28
Diet in the past 12 months	1538	11.74	14.42	0.265
Usual quantity of non-alcohol drinks per day, mean (SD) L	1466	1.75 (0.67)	1.62 (0.62)	0.01
Self-estimated salt consumption				<0.001
Low		30.30	46.01	
Medium		58.28	51.64	
High		11.42	2.35	
Number of days/week consuming meat (%)	1541			<0.001
0–1		13.29	26.73	
2–3		34.82	37.33	
4–5		33.53	26.27	
6 or more		18.35	9.68	
Number of days/week consuming fish (%)	1539			0.002
Less than 1		41.71	53.67	
1		37.32	33.03	
2 or more		20.97	13.30	
Number of minutes walked per day, median (IQR)	1513	30 (20–60)	30 (20–60)	0.20 *
Physical activity at work (%)	1483			0.65
Mainly sitting		5.89	3.83	
Not carrying loads		18.45	18.66	
Carrying light loads		36.89	39.23	
Carrying heavy loads		38.78	38.28	
Physical activity in general	1507			0.61
Never		16.68	20.28	
Less than once per week		14.90	13.2	
Once per week		19.23	18.87	
More than once per week		49.19	47.64	

* test for median, 3 T test.

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
