# Peer review of "Clinical Profiles and Factors Associated with a Low Sodium Intake in the Population: An Analysis of the Swiss Survey on Salt"

_nutrients, 2020, doi:10.3390/nu12113591_

Round 1

Reviewer 1 Report

The paper by Burnier, Pacccaud, and Bochud examined the previously performed Swiss Survey on Salt for behavioral, socio-demographic and clinical profile to determine if they could find evidence of any of these factors that might explain the increased cardiovascular risk associated with a low salt diet. the authors did not measure a difference in the awareness level of the risks associated with salt in individuals consuming either low or high salt.  they did find that low salt consumption was most prevalent in single individuals who were more in charge of their diet and were eating less meat and fish. 

Major criticisms: While there are a number of studies that demonstrate a linear relationship between increasing salt consumption and cardiovascular events, other studies also conclusively demonstrate a "J" shaped relationship with increasing cardiovascular events at both ends of the salt consumption spectrum.

The authors should refer to studies of salt sensitivity that have demonstrated that there are a group of individuals in each study that show an increase in blood pressure while on a low salt diet, which are often referred to as inverse salt sensitive (ISS)(PMID 23197156, 27503851). While studies of cardio vascular disease in ISS have not been published the increase in blood pressure on a low salt diet can exceed established blood pressure guidelines and are expected to result in increased cardiovascular events in those who remain on long term low salt.  The authors should mention the caveat that including individuals who are defined as hypertensive BP>140 mm Hg and on medications might have influenced the outcome of their studies since a large number of these individuals may be ISS (prevalence of ~15%). 

Criticism of the O'Donnell study citing malnutrition as the possible cause of his J shaped relationship to cardiovascular event and salt consumption should also include the probability that some of these individuals may also be ISS.

A citation should be provided that recommends that 3 24 hour urine specimens are more accurate in assessing sodium excretion.

Minor criticisms:

  1. Indirect potentiometry has decreased sensitivity compared to flame photometry and might have influenced the outcome of their studies.
  2. Figure 2 appears to be missing the Men label.

Minor criticisms: The term "limited convenience sampling" should be defined.

Author Response

I would like to thank the reviewer for his/her constructive comments on our paper.

The specficic answers are the following:

1.  The potential impact of an inverse salt-sensitivity is a very interesting observation that may explain the J-shape curve observed in some epidemiological papers and perhaps in our low sodium intake group. Therefore we have included the concept in the introduction and in the limitations of our study. Yet, it has to be mentioned that we do not have much information on the long-term effects of a low salt intake on blood pressure as inverse salt-sensitivity has been reported mainly in short-term protocols. Thus, the impact of inverse salt sensitivity on CV outcomes is not yet well described.

2. A citation has been provided for the recommendation of doing at least three non-consecutive 24h urine collections to assess sodium intake in populations. The reference is : Campbell et al, J Clin Hypertension 2019.

3. Authors agree that potentiometry is less sensitive that flame photometry for the measurement of sodium. However, flame photometry has been abandonned in large University Hospital clinical laboratories for efficiency purposes as large automates are used.

4. The figure as been modified as suggested.

With our best thanks

Reviewer 2 Report

Paper target a very interesting topic of potential negative cardio-vascular effects of low salt intakes, suggested by some observational studies. This topic is quite controversial, as majority of the available evidence support contrary observation. Authors used previously described Swiss Survey on Salt 2010-2012 dataset to examine the differences in subjects with lower and higher salt intakes. Study provide interesting results, that have not been reported previously. The original sampling was extensive, with sufficient power and methodology. Of the limitations is single day sampling, but also our experience shows that even this is very challenging and resulting in quite low participation rates. Authors also reported quite low participation rates (10%), but this is not unusual for studies with 24-h urine sampling. Study seems to be sufficiently controlled to identify samples of non-24h-h urine, although intake of external marker would present even better control (see further comment below). Paper is very well written and would only need minor modifications:

  1. Authors discuss that “subjects eating less (?than? – is there a typo?) 5 g salt per day have a smaller urine volume, suggesting lower fluid intake” and mentioning that this finding is consistent across analyses. On other hand I noted that reported consumption of drinks (Table 4) is significantly higher in this same group (p=0.01). It the above mentioned observation really consistent? This should be at least noted in the discussion, possibly in limitations section. Could it happen that this much smaller group with low sodium intake had more incomplete 24-h urines samples, in comparison to other participants (although not to the extend that participants would be excluded considering urine creatinine excretion criteria). If this is the case, this would also affect estimated low protein intakes in this group of course.  
  2. There are some typos. It is difficult to point those out, as manuscript is not line-numbered, so I suggest that authors go through the text once again carefully (P4: prevalence was described proportions,…)
  3. Supplementary materials should be separated from the manuscript, but it is logical to do this after peer review process.

Author Response

First we would like to thank the reviewer for the positive comments on our paper and the constructive questions and remarks.

The specific answers are the following:

1 Regarding the urinary volume, our data indeed show a consistent increase in urinary volume in subjects eating more than 5 g of salt per day. This is found in the entire studied population and in the comparison of women only. We do not see exactly where a possible contradiction occurs. In table 4,  the low salt eaters have a significant lower number of cups containing caffeine than those eating >5g NaCl per day. The number of drinks is also lower in subjects with < 5g NaCl per day that is 1.62 versus 1.75 . So, in our opinion, this seems consistent.

The point on the completeness of urine collections and the possibility of low quality sampling in subjects with a low urinary sodium excretion is crucial. We do not have better criteria to exclude subjects than those described in the methods section based on urinary creatinine excretion and extremely high or low urinary volume

2. We have tried to correct most typos we found

3. The supplementary tables and figures are now in a separate document.

4. Limited convenience sampling has been defined

Thank you very much.